# Energetic Implications of Morphological Changes between Fish Larval and Juvenile Stages Using Geometric Morphometrics of Body Shape

**DOI:** 10.3390/ani13030370

**Published:** 2023-01-21

**Authors:** Lorena Martinez-Leiva, José M. Landeira, Effrosyni Fatira, Javier Díaz-Pérez, Santiago Hernández-León, Javier Roo, Víctor M. Tuset

**Affiliations:** 1Unidad Asociada ULPGC-CSIC, Instituto de Oceanografía y Cambio Global (IOCAG), Universidad de Las Palmas de Gran Canaria, 35214 Telde, Canary Islands, Spain; 2Instituto Universitario ECOAQUA, Universidad de Las Palmas de Gran Canaria, 35214 Telde, Canary Islands, Spain

**Keywords:** geometric morphometric analysis, ontogenetic development, respiration rate, fishes

## Abstract

**Simple Summary:**

This study aims to assess the relationship between morphology and metabolism during the ontogenetic development of *Chelon auratus*. The geometric morphometric analysis allowed us to identify morphological variations in the transformation from larvae to juveniles and to establish the growth patterns of each stage. According to our results, the ETS activity is linked to the individual morphology, specifically to the body size and to the posterior area.

**Abstract:**

The fish body shape is a key factor that influences multiple traits such as swimming, foraging, mating, migrations, and predator avoidance. The present study describes the body morphological changes and the growth trajectories during the transformation from 24 to 54 days post-hatching in the golden grey mullet, *Chelon auratus*, using geometric morphometric analysis (GMA). The results revealed a decrease in morphological variability (i.e., morphological disparity) with the somatic growth. The main changes affected head size, elongation, and widening of the body. Given that this variability could affect the metabolism, some individuals with different morphologies and in different ontogenetic developmental stages were selected to estimate their potential respiration rate using the Electron Transport System (ETS) analysis. Differences were detected depending on the developmental stage, and being significantly smaller after 54 days post-hatching. Finally, a multivariate linear regression indicated that the specific ETS activity was partially related to the fish length and body shape. Thus, our findings emphasized the relevance of larval morphological variability for understanding the physiological processes that occur during the development.

## 1. Introduction

Theoretical morphology is a scientific discipline arising from the early monographs of the 20th century on the ‘form, shape, and function’ of animal morphologies by Russell [1] and Thomson [2]. Thomson’s work, based on the ideas of Galileo and Goethe on morphology and of Russell on functionalism, was the first to postulate that physical forces and internal growth parameters regulate biological forms and could be revealed via geometric transformations in morphological space [3]. Since then, the theoretical morphology has demonstrated that organism shape is an expression of ecological [4,5,6], evolutionary [7,8], and phylogenetic processes [9,10].

In fishes, the inter- and intraspecific morphological variability is mainly reflected in the body and head shape. Body shape is related to multiple vital activities, such as swimming, searching for food, evading predators, courtship dances, and territory defence [11,12,13], whereas head shape is mainly linked with foraging and prey selection [14,15,16]. The fish shape has a high phenotypic plasticity: the ability of individual genotypes to produce different phenotypes when exposed to different environmental conditions [17,18,19]. It is conspicuous from the earliest stages of ontogenetic development [20,21]. The plasticity is considered as an ecological strategy to ensure the survival of the species [22,23], which has great relevance during the spawning and recruitment periods [24], known as “critical periods” [25]. Ecological traits expressed by juveniles and adults can be affected by environmental factors during embryo development through epigenetic modifications [26,27] or morphological alterations (e.g., number of vertebrae and muscle fibres) [28]. Moreover, fish larvae have many critical periods that are correlated with ontogenetic events (e.g., preflexion and flexion), which are affected by environment, leading to phenotypic response [29,30,31]. In this sense, morphological, sensorial, and behavioral changes occur throughout the fish ontogenetic development depending on exogenous factors (e.g., food supply and temperature) and on physiological characteristics (e.g., type of respiration and muscle reorganization) that affect the metabolism of each specimen [21,32,33]. During the ontogenetic development, somatic growth tends to be allometric with a high rate during the larval stage, while it is isometric and slower during the early juvenile stage [34,35] and stabilizes thereafter [36]. These metabolic changes are related to the development of vital organs and sensory mechanisms that optimize their survival [37,38,39]. Thus, body shape and metabolic rate are coupled in early life stages of fish [40], understanding the shape as body mass [41,42]. However, morphological changes are clearly visible, such as the position and size of the eyes (which affect visual capacity), the size of the mouth (which influences ingestion capacity), the area of the gill apparatus (which increases the efficiency of respiration), and the pattern of the caudal region (which relates to locomotion) [20,21,34,37].

Distance measurements and geometric morphological analysis (GMA) are the common tools for assessing the degree of change in shape, and the latter is the most powerful for describing different visual patterns [43,44]. In particular, GMA has been used in the early ontogenetic fish stages to evaluate the phenotypic flexibility in different environmental conditions (e.g., temperature, salinity, and pH) [31,45]. In addition, GMA has been used to influence ecological factors (e.g., feeding and preference of habitat) [46] to visualize ontogenetic changes [21,47,48] and to identify deformities [49,50]. Recently, González and Nicieza [51] have used the GMA to explain the effects of ontogeny and size, on the link between shape and metabolic scaling in one- and two-year-old juvenile brown trout. In general, all studies concluded that a more streamlined body shape displayed maximum metabolic rates than a deep-bodied species at intra- and interspecific levels [51,52,53], which influenced the prolonged swimming capability [54]. However, there is an important gap between the body shape and the metabolism in these stages, which may be crucial to evaluate metabolic scaling theories.

The present study is the first approach to apply GMA for exploring the morphological changes during the early stages of ontogenetic development and its relevance with the metabolic activity. The main goals are (i) to display the change in body shape that occurs during ontogenetic development, (ii) to estimate the relative metabolic activity of specimens with different morphology and days post-hatching, and (iii) to evaluate the effect of fish size and body shape on the potential respiration of individuals. For our purpose, we selected specimens of *Chelon auratus* (Risso, 1810) (Pisces: Mugilidae) because it is an important species for aquaculture due to their euryhaline and eurythermal adaptability, which facilities their farming [55]. In fact, there are industrial (extensive and semi-extensive) aquaculture activities in the Mediterranean region using ponds and reservoirs, which mainly focus on nutrition [56]. Furthermore, physiological and morphological studies in these initial stages have already been performed in other mugilids species [36,39], which could help us in the interpretation of our findings.

## 2. Materials and Methods

### 2.1. Study Sampling

Specimens were obtained from the Scientific and Technologic Park of the University of Las Palmas de Gran Canaria (Las Palmas, Canary Island, Spain). This species shows the absence of cannibalism, avoiding the loss of morphological and physiological variability within tanks. Due to limitations to obtain larvae, we studied two developmental stages. Larvae were acclimatized in tanks with temperatures ranging between 19 and 20 °C, and the oxygen concentration was around 6–7 mg·L^−1^. During the first 17 days post-hatching (dph), larvae fed on the rotifer *Brachionus plicatilis*, and on the artemia nauplii *Artemia salina*, both enriched. Thereafter, larvae fed on commercial pellets.

For our experimental study, we set two sampling times for fish at 24 and 54 dph. The criterion for selecting these periods was based on previous studies in species of Mugilidae, where the differentiation between larvae and the juvenile stage was established around 40–60 days [57,58,59]. Specimens were captured with a small net and euthanized with tricaine solution for 5–10 s. A total of 131 specimens from 24 dph were collected and photographed using a camera attached to the stereomicroscope (Leica, EZ4W, Wetzlar, Germany). One hundred eighteen individuals obtained after 54 dph were photographed using a digital camera (Nikon, D70, Tokyo, Japan) due to their large size. After imaging, all specimens were labelled and stored in liquid nitrogen for 2 h and later preserved at −80 °C until the metabolic analysis.

### 2.2. Geometric Morphometric Analysis

To characterize the morphology of body shape, we used GMA based on the landmarks-based method [43,60]. This method consists of a set of two- or three-dimensional landmark coordinates to record the geometry of the structure to evaluate the degree of change [44]. The scheme of landmarks (fixed homologous points) and semi-landmarks (sliding or mobile non-homologous points), consisting of 18 different points defining the general and head shape (Figure 1), was selected using common configurations to previous studies [51,61]. Some individuals were discarded due to alterations in the position.

Digitalization of these points was performed using the software tpsDig v. 1.81 [34], as well as the measure of the standard length (*SL* in mm) (Figure 1). tpsSmall 1.28 software package [62] was used to evaluate the approximation of the distribution of the specimens in Kendall’s shape space relative to the linear tangent space for each analyzed view [63]. The correlation coefficient between tangent distances and the Procrustes distances was high (uncentered correlation = 1, root mean square error = 0.000003), indicating that the amount of shape variation was small enough to permit statistical analyses using only the Procrustes distances. A generalized Procrustes analysis (GPA) was performed [63,64,65] on the raw landmarks data to translate specimens to a common location in coordinate space, scale, and rotate them to reduce the distances between homologous landmarks. The analysis was performed using the package *geomorph* v. 4.04 [66,67] in R environment (R Development Core Team, 2022). A scale was included in the images to allow the acquisition of a scaling factor for calculating centroid sizes (*CS*), which was defined as the square root of the summed squared interlandmark distances [68]. In fact, log-*CS* was highly correlated with log fish length (log-*SL*) of the specimens (*r*^2^ = 0.799, *p* = 0.001, 9999 permutations). Principal Components Analysis (PCA) based on the variance–covariance matrix of the aligned specimens was performed to describe how shape varied between stages. Significant eigenvectors were identified by plotting the percentage of total variation explained by the eigenvectors versus the proportion of variance expected under the “broken-stick model” [69]. Thin-plate spline deformation grids showing shape variation along the PC axes [68] were constructed with the PAST software v. 4.03 [70].

### 2.3. Electron Transport System (ETS) and Protein Measurements

To study the potential respiration during ontogeny development, we analyzed the *ETS* activity method. A subsample was taken according to morphological results obtained. A total of 58 individuals were selected (Appendix A), some of them were distributed in key positions in the morphospace (ends of axes), and the others were taken random. The *ETS* activity was measured following the method of Packard et al. [71] modified by Gómez et al. [72]. The first step consisted of homogenization of each sample in a phosphate buffer using an electric homogenizer and they were centrifugated for 10 min at 4000 rpm at 0–4 °C. Thereafter, a subsample was taken of the liquid phase resulting from centrifugation and incubated at 18 °C and mixed with a buffer reaction containing NADH and NADPH coenzymes, succinate, and tetrasodium salt (INT, artificial electron acceptor). The reaction was stopped after 20 min of adding quench solution. This entire procedure was performed at a low temperature to avoid degradation of enzyme activity and protein, and for each sample, a blank assay was made without *ETS* substrates. Finally, using a spectrophotometer, we measured the *ETS* activity at 490 and 750 nm.

To determine biomass in terms of protein, we used the previous subsample following the method described by Lowry et al. [73] with modifications of Rutter [74]. We used bovine serum albumin (BSA) as the standard, and we measured at 750 nm in the spectrophotometer. Specific *ETS* activity (μlO_2_·mg protein^−1^ h^−1^) was used to characterize the potential respiration rate of each specimen. Assuming an activation energy of 15 kcal·mol^−1^ [71], a quite conservative respiration I to *ETS* (R/*ETS*) ratio of 0.5 was used as in zooplankton [72].

### 2.4. Statistical Analysis

To detect differences for the set of Procrustes shape variables between larval and juvenile stages, an analysis of variance (ANOVA) was carried out considering the ’stage’ and ‘centroid size’. To quantify the phenotypic variability between stages, we calculated the morphological disparity (MD) using the function *morphol. disparity* in geomorph. This measure was estimated as the Procrustes variance for groups, using residuals of a linear model fit [75]. The PC1 and PC2 were plotted to build the morphospace for explaining the variability of the fish body shape [76]. Finally, a multiple linear stepwise regression was used to predict the relationship between log-*ETS* and log-*CS* and PC components with the function *stepAIC* in the package *MASS* v. 7.3–58.1 [77]. Previously, we tested the multicollinearity and homoscedasticity between variables using the Spearman correlation (non-normal data) and Goldfeld–Quandt test, respectively. The mean level of ETS between stages (24 *versus* 54 dph) was compared using a Student *t*-test (permutations = 9999). The assumption of the normality and homogeneity of variance were previously checked using Shapiro–Wilk and *F* tests [78].

## 3. Results

### 3.1. Body Analysis

A total of 110 and 116 individuals were collected during the larval and juvenile stages, respectively (Appendix A). The SL increased with the ontogenetic development, hence larval individuals ranging from 6.7 to 15.2 mm, and juveniles from 14.5 to 38.8 mm. The ANOVA analysis showed significant differences in the body shape (*F* = 25.11, *p* < 0.001) and centroid size (*F* = 215.52, *p* < 0.001) between the development stages (Table 1). The morphological disparity significantly differed between stages, which was double in the larval stage (*MD*_L_ = 0.00222 vs. *MD*_J_ = 0.00158, permutations = 9999, *p* < 0.001), demonstrating a higher morphological heterogeneity in this development phase.

The first three components of the PCA analysis accounted for higher variance than expected by chance alone (78.8%), reaching the first two components 70.5% of variance. The morphospace illustration (PC1 *versus* PC2 and, PC1 *versus* PC3) showed a clear ontogenetic difference between the stages studied (Figure 2A,B). The PC1 explained 54.4% of variance, the positive values identified specimens with a higher proportion of head and big eyes (larvae stage); whereas the negative values showed individuals with wider and shorter heads and a more elongated body shape (juvenile stage). The PC2 attained 16.1% of variance and was linked to fish body variation from the anal fin position to tail ending. The positive-axis represented individuals with a shorter body, while the negative-axis illustrated specimens with an enlarged development. Contrary to PC1, PC2 equally affected larvae and juvenile stages (Figure 2A). The PC3 (8.3% of variance) was linked to the height and elongation of body shape. The positive-axis represented individuals with a higher dorsal widening fish body and greater proportion of the region between the anal and caudal fins, while the negative-axis illustrated specimens with a greater ventral development fish body and lesser development of the posterior region. Like PC2, the point distribution of larvae and juveniles did not differ along the PC3 axis (Figure 2B).

Correlation between PC components and log-centroid size (log-*CS*) were significant for PC1 (*p* < 0.001), explaining the 82.1% of total variance. When the analysis was performed separately for each stage, significant differences were obtained between slopes of linear regressions (mean ± standard error, *b*_L_ = 0.078 ± 0.004 for larvae and *b*_J_ = 0.024 ± 0.004 for juveniles; *t*-test = 9.134, *p* < 0.001), demonstrating a change in the somatic growth between the stages (Figure 3). Moreover, the shape (i.e., distribution points along PC1 axis) was clearly different between stages, showing a higher heterogenous larvae than juveniles. The PC2 component scarcely explained 2.9% of variability (*p* = 0.012) and it was not in correlation with PC3 (*r* = 0.077, *p* = 0.251).

### 3.2. Metabolic Analysis

The *ETS* values were normally distributed for larval and juvenile specimens (Shapiro–Wilk test, *W* = 0.969, *p* = 0.760; *W* = 0.978, *p* = 0.776, respectively). The *F*-test (*F* = 7.605, *p* < 0.001) showed significant differences between their variances (2.63 and 20.01 for larval and juvenile stage, respectively). The comparison of mean values between stages provided significant differences between the stages (*t*-test, *t* = 11.174, *p* < 0.001).

The linear relationships between all variables indicated a weak correlation of PC1 (*r* = −0.290, *p* < 0.05) with the log-*ETS*, whereas it was stronger with PC2 (*r* = −0.860, *p* < 0.001) (Figure 4). Consequently, the linear stepwise regression analysis was only performed using PC1 and PC2. These two variables explained the 23% of total variance of *ETS* activity (*F*_2,47_ = 7.02, *p* = 0.002), mainly indicating that smaller individuals and a greater anal–caudal region had a higher metabolic activity.

## 4. Discussion

In recent years, many studies have focused on embryonic and larval development for the domestication of mugilids as important target species for aquaculture [79,80,81]. Among other reasons, they are low trophic level species, which makes them undemanding in the type of prey to ingest, their organ development is fast, and they show a good adaptability to hatching in different environmental conditions, mainly regarding salinity and temperature [55,82]. However, despite significant advances, rearing conditions and morphological changes during larval development are still under investigation [59,83]. In this context, our study describes the common morphological changes that occur during the ontogenetic development from larval to juvenile stage in fishes. These changes are visible in the eye position and size, mouth size, gill apparatus area, and body elongation especially in the anal–caudal region. They were not uniform during the growth for the whole larvae, which explains the high grade of morphological heterogeneity. From an ecological perspective, the morphological disparity reflects a fast allometric growth [34,35]. This could potentially mean that larvae exhibit an adaptive suitability to the different physio-chemical environment conditions [84] driving their recruitment, distribution, and survival [85]. The larval stage finishes when the disparity decreases and the growth becomes isometric, which is an omen of the onset of the juvenile stage [35,39]. Throughout this transition, body shape of larvae changes to become deeper and laterally compressed, which is more appropriate for fast swimming [45,86,87]. The shift in shape can be interpreted as a transition from anguilliform to subcarangiform motion [39,88,89].

The estimation of oxygen consumption is important in energy budgets, especially when attempting to determine energy requirements for growth and survival of fish larvae [90]. Studies have concluded that specific *ETS* activity increases with the ontogenetic development from eyed egg to juvenile stages [91,92] present study. Future climate change conditions seem to increase the energy demand during the embryonic and larval stages, which potentially may constitute a survival bottleneck [92,93]. Knowing that the metabolic activity and growth rate are interconnected [94,95], different studies have focused on the intensity of protein degradation during the active growth period. The studies demonstrated its utilization as an energy source for swimming, reducing the locomotion cost for obtaining a higher speed [88,96,97]. Khemis et al. [39] observed in *Chelon labrosus* that during metamorphosis, the fish developed an adult axial muscle distribution, showing a decrease in the red muscle (aerobic respiration) and an increase in the white muscle in this area (anaerobic respiration) in the anal–caudal region. This region is key for body movement during swimming and requires high protein content and oxygen supply. This organization and development of the axial musculature was related to the increased swimming performance during the larval stage to reduce predation. Although the present study did not analyze the musculature organization between stages, we found that the anal–caudal region varied independently of the ontogenetic development (PC2 component) and showed a weak correlation with *ETS* activity. Considering that high levels of ETS require greater muscle mass and cell respiration [87], proportion of the muscular types vary during the transformation of larval to juvenile individuals [39,98]. It is plausible to think that there is a tridimensional growth favoring the anaerobic respiration and a high activity, independent of ontogenetic stage. Given the limitations of geometric morphometric analysis used in the present study, all evidence suggests that individual growth in body shape elongation may be more interesting than the ontogenetic stage itself. The results can contribute to understanding the energetic requirements of individuals during these early vulnerable stages.

From an eco-evolutionary point of view, the observed phenotypic plasticity may be linked with the ability of mugilids to cope well with salinity fluctuations shortly after hatching until the larva stage [99,100,101]. However, oxygen consumption does not appear to vary in eggs and larvae reared at different salinities [101]. This could likely be due to the osmoregulatory mechanisms developing precociously [102]. In addition, individuals with higher *ETS* activity and more developed anal–caudal regions should have higher swimming performance. These individuals could be more efficient in avoiding predators and feeding [103], which will eventually favor their survival and growth, respectively. However, observational experiments are needed to demonstrate this.

## 5. Conclusions

The present study represents a first step to assess the body shape changes during ontogenetic development and their relationship with fish metabolism. We found high morphological disparity during ontogenetic development, even under constant environmental conditions. Different growth trajectories and morphological variability were observed during the development, although the morphological shifts in the anal–caudal region could be related to variations in the type and disposition of muscle and energy demands. Given that this morpho-functional variation is independent of ontogenetic development, the present study opens new lines of research needing to be clarified, especially under different scenarios of climatic changes where the phenotypic plasticity can play a key role in the sustainability of fishery resources.

## Figures and Tables

**Figure 1 animals-13-00370-f001:**
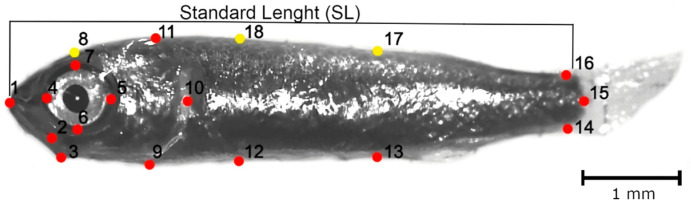
Location of selected landmarks (red dots) and semilandmarks (yellow dots) to describe body shape in C. auratus specimens of 24 dph. L1, anterior tip of the premaxilla; L2, posterior tip of the premaxilla; L3, angle of the lower jaw; L4, anterior margin in the eye; L5, posterior margin in the eye; L6, inferior margin in the eye; L7, superior margin in the eye; L8, dorsal margin of the head; L9, ventral margin in the end of the head; L10, posterior margin in the end of the head; L11, dorsal margin in the end of the head; L12, insertion of the pelvic fin; L13, anterior insertion of the anal fin; L14, ventral insertion of the caudal fin; L15, central of the caudal fin; L16, dorsal insertion of the caudal fin; L17, dorsal projection of the landmark 13; L18, dorsal projection of the landmark 12.

**Figure 2 animals-13-00370-f002:**
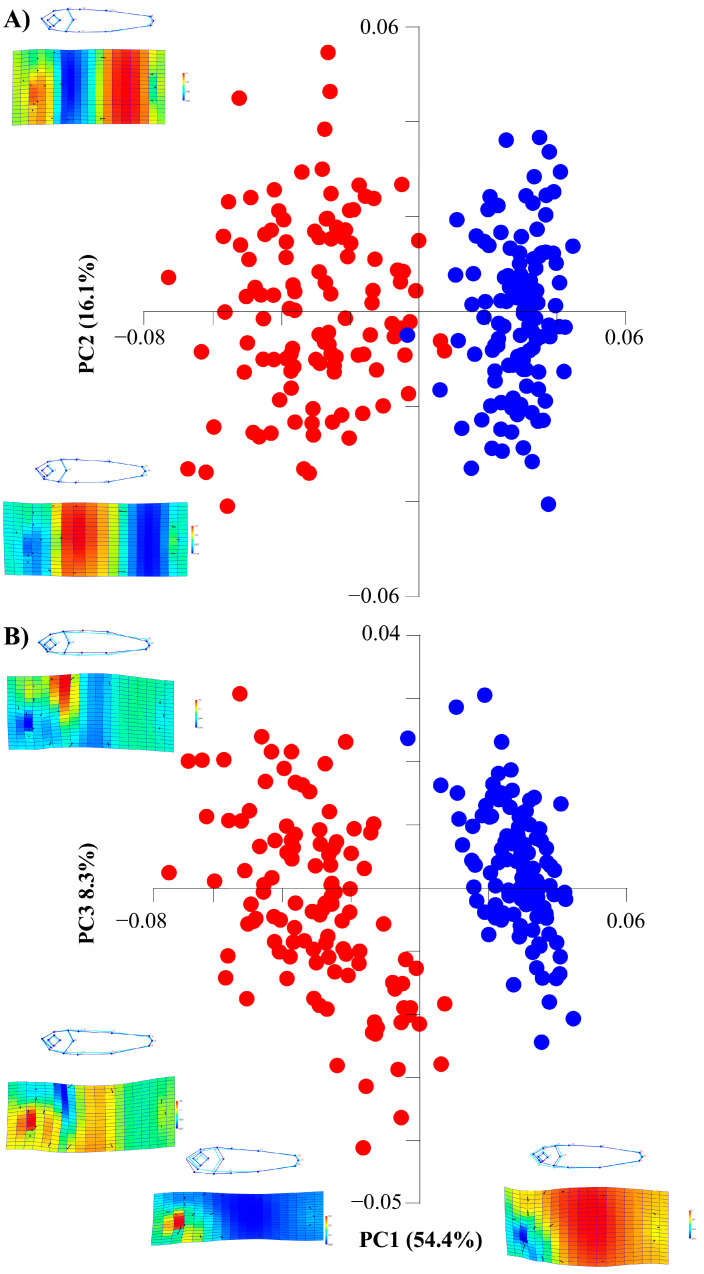
Morphospace illustrations between PC1-PC2 (**A**) and PC1- PC3 (**B**) for detecting body shape variations during larval and juvenile stages of *Chelon auratus*. Red dots are larvae (24 dph), blue dots are juveniles (54 dph). Thin-plate spline deformation grid displays body shape ranges; expanded shape is represented with warm colors and contracted shape in cold colors.

**Figure 3 animals-13-00370-f003:**
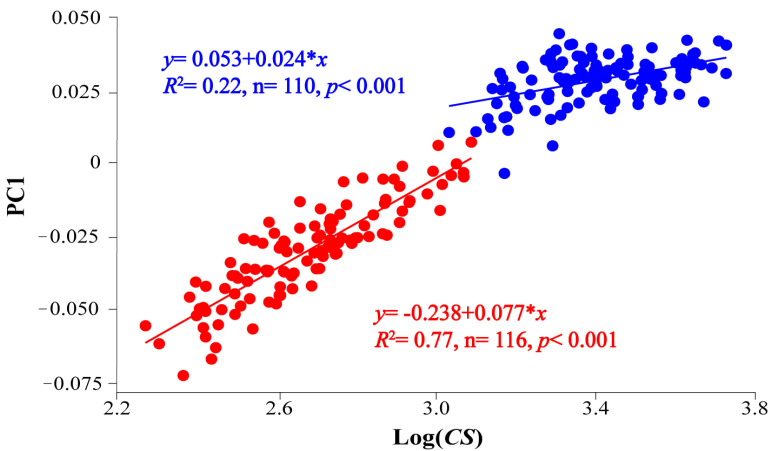
Linear regression between PC1 and log-centroid size (log-*CS*) during larval and juvenile stages of *Chelon auratus*. Red dots are larvae (24 dph), blue dots are juveniles (54 dph).

**Figure 4 animals-13-00370-f004:**
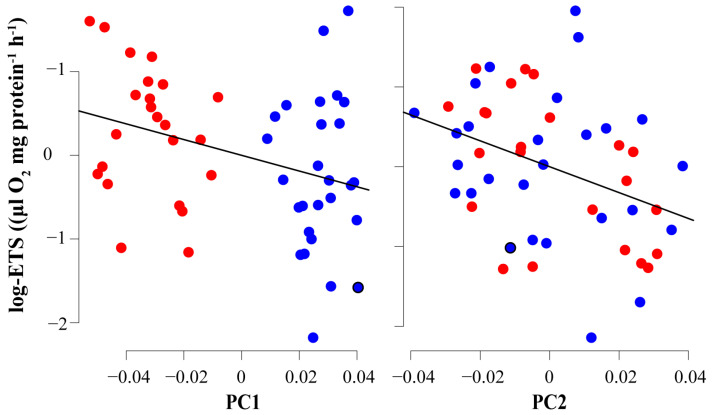
Stepwise regression analysis between the metabolic activity (log-*ETS*) and significant PC components during larval and juvenile stages of *Chelon auratus*. Red dots are larvae (24 dph), blue dots are juveniles (54 dph).

**Table 1 animals-13-00370-t001:** ANOVA results to determine the effect of centroid size and ontogeny phase on the body shape of *Chelon auratus*.

	*df*	SS	MS	Rsq	*F*	*p*
Centroid	1	0.199	0.199	0.465	215.52	0.001
Stages	1	0.023	0.023	0.054	25.11	0.001
Residuals	223	0.206	0.0009	0.481		
Total	225	0.428				

*df*: degrees of freedom; *F*: test statistic; MS: mean of sum squares; Rsq: percentage of variation; SS: sum of squares; *p*-value significance.

## Data Availability

The data presented in this study are available in the article and Appendix A.

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
