# Peer review of "Energetic Implications of Morphological Changes between Fish Larval and Juvenile Stages Using Geometric Morphometrics of Body Shape"

_animals, 2023, doi:10.3390/ani13030370_

Round 1

Reviewer 1 Report

Line 74 - rotifers and Artemia ,enriched ? or non-enriched? pls indicate.

Line 82 - Microscope model and country, pls indicate. Same for camare model at line 83.

Author Response

Line 74 - rotifers and Artemia ,enriched ? or non-enriched? pls indicate.

It has been indicated in the line 75 (now line 82). as follows: "...during the first 17 days post hatching (dph), larvae fed on the rotifer Brachionus plicatilis, and on the artemia nauplii Artemia salina, both enriched."

Line 82 - Microscope model and country, pls indicate. Same for camare model at line 83. 

We have added the information as follows: "A total of 131 specimens from 24 dph were collected and photographed using a camera attached to the stereomicroscope (Leica, EZ4W, Germany). One hundred eighteen individuals obtained after 54 dph were photographed using a digital camera (Nikon, D70, Japan) due to their large size."

Now lines 88 and 89.

The English have been newly revised.

Reviewer 2 Report

This paper reported the morphological changes and ETS activity between the larval and juvenile stages of fish. I found it quite interesting and valuable. It can be accepted for publishing after some minor revisions. I have only comments on the background introduction and discussion.

General comments:

Introduction. Why was the species Chelon auratus chosen as an experimental animal? Are there any previous studies of this fish, i.e. metabolism, morphology, and growth?  Some necessary background should be added.

How could you be sure that all individuals of 24 dph were larvae while all individuals of 54 dph were juveniles? Even though this species may generally be juveniles at 40-60 days as in the literature, it may not be the case for all individuals. Figure 3 also showed an obvious transition within the juveniles, which may be more important than the difference between larvae and juveniles. Therefore, both growth and ontogeny affect the morphological changes. An anatomical test would be helpful distinguish growth and ontogeny. However, my major point is not questioning the design or terminology, on the contrary, I found it may be an interesting issue to be discussed. The results showed both the fish of 24 dph and 54 dph showed large within-group variation in body length and some larvae actually grew to similar sizes as the juveniles. Could the within-group variation be majorly due to growth while the between-group difference is majorly due to ontogeny? A multiple regression (maybe a segmented line model?) including both CS and ontogeny stage as independent variables could be a better way than that in Figure 3.

Minor comments

L42. understanding the shape as body mass. What does it mean?

L166. Does the body shape mean PC1?

Fig. 3. It seems an inflection within those juveniles, suggests a diversity. Does any potential ontogenetic change happen for those 54 dph fish?

L216. What was the correlation between log-ES and log CS? It could supply evidence for metabolic scaling studies.

Fig. 4. The within-group correlation could also be analyzed.

Author Response

Introduction. Why was the species Chelon auratus chosen as an experimental animal? Are there any previous studies of this fish, i.e. metabolism, morphology, and growth?  Some necessary background should be added.

Thanks you for your comment. We have added to the end of the Introduction section a paragraph justifying why this species was selected and its relevance on the context of the aquaculture. The text is as follows: "...The main goals are (i) to display the change in body shape that occurs during ontogenetic development (ii) to estimate the relative metabolic activity of specimens with different morphology and days post hatching, and (iii) to evaluate the effect of fish size and body shape on the potential respiration of individuals. For our purpose, we selected specimens of Chelon auratus (Risso, 1810) (Pisces: Mugilidae) because it is an important species for aquaculture due to their euryhaline and eurythermal adaptability, which facilities their farming [25]. In fact, there are industrial (extensive and semi-extensive) aquaculture activities in the Mediterranean region using ponds and reservoirs, which mainly focus on nutrition [26]. Furthermore, physiological and morphological studies in these initial stages have been already performed in other mugillids species [6, 10], which could help us in the interpretation of our findings."

New references have been added and listed.

How could you be sure that all individuals of 24 dph were larvae while all individuals of 54 dph were juveniles?

This was not a really problem because the head morphology was clearly different between both stages. Certainly, we have doubts in five individuals (3 larvae and 2 juveniles) but the elimination of this specimens would lead to irreal scenario of the ontogenetic development. Besides, considering the literature about the transition between larvae and juveniles in other other mugillids, we think that the our criterion is acceptable. In fact, these cases do not correspond with individuals with special ETS.     

Could the within-group variation be majorly due to growth while the between-group difference is majorly due to ontogeny? A multiple regression (maybe a segmented line model?) including both CS and ontogeny stage as independent variables could be a better way than that in Figure 3.

Both the somatic growth and ontogeny influence in the intra and inter-variability. We have modified the text for better understanding by the reader as follows: "When the analysis was performed separately for each stage, significant differences were obtained between slopes of linear regressions (mean ± standard error, bL= 0.078 ± 0.004 for larvae and bJ= 0.024 ± 0.004 for juveniles; t-test= 9.134, p< 0.001) demonstrating a change in the somatic growth between the stages (Figure 3). Besides, the shape (i.e., distribution points along PC1 axis) was clearly different between stages, showing a higher heterogenous larvae than juveniles. The PC2 component ....".

We do not think necessary to make the mathematical procedure proposed by the reviewer since it would not improve the interpretation of results. Consequently, we would like to kept the Figure 3 if the reviewer is satisfied with the proposed modification of the text.

L42. “understanding the shape as body mass”. What does it mean?

The changes in the shape are usually established from GMA, but in physiological studies the 'volumen' is more important than a bidimensional analysis. For that reason, they use the 'body weight' as a simplification 3D of shape, taking into account that fish length and weight are correlated. Questionable, but real. 

L166. Does the body shape mean PC1?

No, here we only analyzed the Procrustes data. 

Fig. 3. It seems an inflection within those juveniles, suggests a diversity. Does any potential ontogenetic change happen for those 54 dph fish?

Of course. This is more evident in Khemis et al. (2013, see figure 8) for Chelon labrosus. Our specimens will undergo changes after 54 dph, but it was not possible to make a new sampling for demonstraing due to causes unrelated to our interests.

L216. What was the correlation between log-ES and log CS? It could supply evidence for metabolic scaling studies.

It is true. It is implicated in the text. In the line 198, we said:" Correlation between PC components and log-centroid size (log-CS) were significant for PC1 (p< 0.001), explaining the 82.1% of total variance." Therefore, when we established a significant correlation between log-ETS and PC1, we are assertion that ETS is correlated to log-CS. 

Fig. 4. The within-group correlation could also be analyzed.

We have already analyzed this variability. It was not included because it was not significant for both stages. However, this is very interesting for future studies, but we have to increase the number of specimens for stage since the variability is very high. Besides, we did not give this information to avoid more speculation. We are aware that this study is very preliminary and needs to be improved to address some issues such as intra-group variability.